ⓐ | Open Peer Review | Applied and Industrial Microbiology | Research Article

# New myxobacteria of the *Myxococcaceae* clade produce angiolams with antiparasitic activities

Sebastian Walesch,[1,2,3,4] Ronald Garcia,[1,2,3,4] Abdelhalim B. Mahmoud,[1,2,3,4,5] Fabian Panter,[1,2,3,4,6] Sophie Bollenbach,[1,2,3,4] Pascal Mäser,[7,8] Marcel Kaiser,[7,8] Daniel Krug,[1,2,3,4] Rolf Müller[1,2,3,4,6]

**ABSTRACT**  In the past century, microbial natural products have proven themselves to be substantial and fruitful sources of anti-infectives. In addition to the well-studied Actinobacteria, understudied bacterial taxa like the Gram-negative myxobacteria have increasingly gained attention in the ongoing search for novel and biologically active natural products. In the course of a regional sampling campaign to source novel myxobacteria, we recently uncovered new myxobacterial strains MCy12716 and MCy12733 belonging to the *Myxococcaceae* clade. Early bioactivity screens of the bacterial extracts revealed the presence of bioactive natural products that were identified as angiolam A and several novel derivatives. Sequencing of the corresponding producer strains allowed the identification of the angiolam biosynthetic gene cluster, which was verified by targeted gene inactivation. Based on bioinformatic analysis of the biosynthetic gene cluster, a concise biosynthesis model was devised to explain angiolam biosynthesis. Importantly, novel angiolam derivatives uncovered in this study named angiolams B, C, and D were found to display promising antiparasitic activities against the malaria pathogen *Plasmodium falciparum* in the 0.3–0.8 µM range.

**IMPORTANCE**  The COVID-19 pandemic and continuously emerging antimicrobial resistance (AMR) have recently raised awareness about limited treatment options against infectious diseases. However, the shortage of treatment options against protozoal parasitic infections, like malaria, is much more severe, especially for the treatment of so-called neglected tropical diseases. The detection of anti-parasitic bioactivities of angiolams produced by MCy12716 and MCy12733 displays the hidden potential of scarcely studied natural products to have promising biological activities in understudied indications. Furthermore, the improved biological activities of novel angiolam derivatives against *Plasmodium falciparum* and the evaluation of its biosynthesis display the opportunities of the angiolam scaffold on route to treat protozoal parasitic infections as well as possible ways to increase the production of derivatives with improved bioactivities.

**KEYWORDS**  natural products, myxobacteria, drug discovery, anti-parasitic, structure elucidation, biosynthesis

M any infectious diseases have lost their status as life-threatening conditions due to the discovery, development, and application of—often natural products-based —drugs. However, protozoal parasitic infections remain among the most devastating causes of mortality and morbidity, particularly in low- and middle-income countries. These diseases affect more than a billion people worldwide and are contributing to poverty and under-development (1–3). Examples from the WHO list of neglected tropical diseases (NTDs) are human African trypanosomiasis caused by *Trypanosoma rhodesiense* spp., Leishmaniasis caused by *Leishmania* spp., and Chagas disease (American

Address correspondence to Rolf Müller, Rolf.Mueller@helmholtz-hips.de.

The authors declare no conflict of interest.

10.1128/spectrum.03689-23 **1**

trypanosomiasis) caused by *Trypanosoma cruzi*. (4) The three pathogens belong to the Trypanosomatidae, a large family of flagellated protozoa. Although malaria, caused by *Plasmodium falciparum*, is no longer considered an NTD since 2000, the disease still remains a major global health challenge due to its heavy death toll and high mortality levels among pregnant women and children in malaria-endemic African countries (5).

The drugs currently available to treat these diseases are problematic given their serious adverse effects, limited efficacy, and most importantly the emergence of drug resistance. Therefore, there is an urgent need for the development of new, efficacious, safe, and cost-effective drugs for the fight against protozoal tropical diseases (6–8).

Natural products remain a successful source of inspiration for the discovery of new drugs. About two-thirds of the drugs launched over the last 20 years derive directly or indirectly from natural resources (9). Moreover, of the 20 approved antiparasitic drugs, 9 were natural products or derivatives thereof (9). So far, all approved natural product-based drugs for the treatment of protozoal infections are plant-derived (9), but microbial natural products can also have promising bioactivities against protozoa, as displayed by fungal leucinostatins (10) or myxobacterial macyranone A (11) or chlorotonil A (12). As demonstrated by the structural diversity and variety of biological functions of the over 32,500 reported natural products, microorganisms are a prolific source of diverse and putatively biologically active metabolites (13). This plethora of natural products holds the promise of hidden bioactivities against often under-screened pathogens. Considering myxobacterial natural products, this is displayed by the recent findings of chondramides active against SARS-CoV-2 (14), the antitubercular properties of myxovalargin (15), or the potent anti-filarial activity of corallopyronin A through the inhibition of *Wolbachia* endosymbionts (16). Moreover, evaluation of the biosynthesis can be used to improve production titers in the heterologous expression of a natural product, as displayed for corallopyronin A (17), or to heterologously produce improved derivatives of a natural product family, as displayed for disorazoles (18).

Following the initial purification of the antifungal ambruticin from a *Sorangium* sp. strain (19), myxobacteria as a group have increasingly gained attention as prolific producers of natural products. Their immense potential to produce compounds with novel chemical scaffolds and a wide range of biological activities is displayed by their large genomes with many biosynthetic gene clusters (BGCs), as well as the over 600 known myxobacterial natural products (20, 21). Taking into account that the majority of myxobacterial taxa has not been isolated yet (22) and that increasing phylogenetic diversity in myxobacteria entails increasing chances to find new chemical scaffolds (23), sampling campaigns to find, isolate and cultivate novel myxobacteria are a promising undertaking.

In this study, we describe the isolation of new myxobacterial producers of natural products leading to the discovery of four novel derivatives of the known natural product angiolam A (24, 25). Whole-genome sequencing of the new strains allowed the identification of the angiolam biosynthetic gene cluster that was verified by gene disruption in the genetically amenable producer *Pyxidicoccus fallax* An d48 (26, 27). Furthermore, we report the sub-micromolar bioactivities of several novel angiolam derivatives against the malaria pathogen *Plasmodium falciparum* that might make it an option for the development of future anti-malarial drugs.

## RESULTS AND DISCUSSION

### New producers of angiolam

As part of our ongoing efforts to isolate, investigate, and cultivate new myxobacterial strains as producers of novel and bioactive natural products, we launched a regional citizen science campaign named "MICROBELIX" (http://www.microbelix.de). In this project, soil samples were taken by members of the general public and channelled into the institute's strain isolation and characterization workflow. Among the isolates from this sampling campaign, strain MCy12716 caught our attention because

its crude extract displayed anti-parasitic activities in an initial screening. Ultra-high performance liquid chromatography-mass spectrometry (UHPLC-MS) analysis of the bacterial extract followed by dereplication of the secondary metabolite profile of MCy12716 revealed decent production of the myxobacterial antibiotic angiolam A (1), previously described from *Pyxidicoccus fallax* An d30 (formerly classified as *Angiococcus disciformis* strain An d30, Fig. 1) (25). In addition, a number of compounds with similar fragmentation patterns suggestive of similar chemical scaffolds were observed (Fig. 2A and B). However, MCy12716 showed strong variations in angiolam production within subsequent cultivations; hence, we searched for an alternative strain showing more stable production patterns. Thus, strain MCy12733 was eventually chosen for further analysis of the extended angiolam compound family. Based on phylogenetic analysis, strains MCy12716 and MCy12733 represent two novel producers of angiolams, which are located in a phylogenetically divergent clade compared to the original producer strains An d30 and An d48 (Fig. 1). The two novel isolates clustered within the *Pyxidicoccus trucidator-Pyxidicoccus caerfyrddinensis-Pyxidicoccus xibeiensis* clade, while the previously known angiolam producers branched with *Pyxidicoccus fallax*. The biosynthetic ability to produce angiolam, therefore, seems to be strongly correlated to the genus *Pyxidicoccus*.

## Identification, production, and structure elucidation of novel angiolam derivatives

In order to obtain a better overview of the natural product repertoire of MCy12733, the strain was cultivated in a standard Cy/H medium, and the resulting extracts were

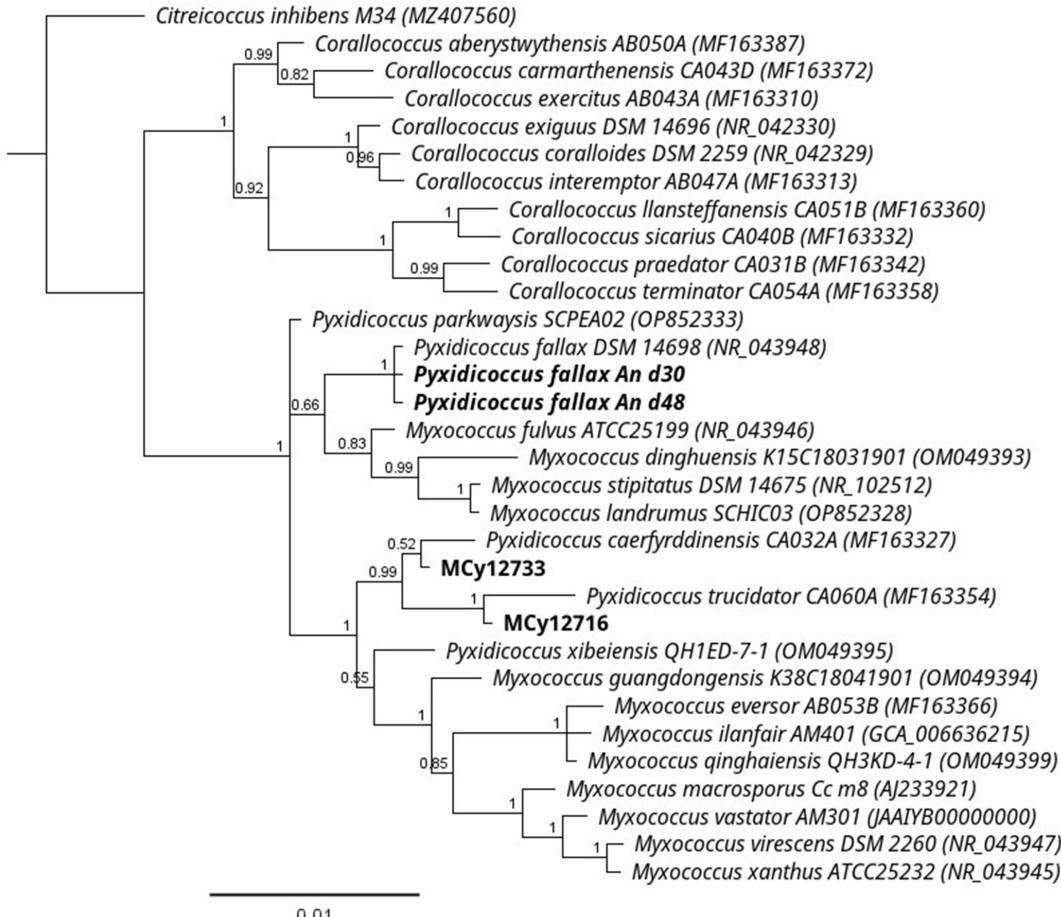

**FIG 1** Phylogenetic tree (MrBayes) inferred from 16S rRNA gene sequence showing the position of angiolam-producing strains (bold) in the *Myxococcus-Pyxidicoccus* clade. The values at nodes represent the posterior probability. The tree is rooted with *Citreicoccus inhibens*. Bar, 0.01 nucleotide substitution per site.

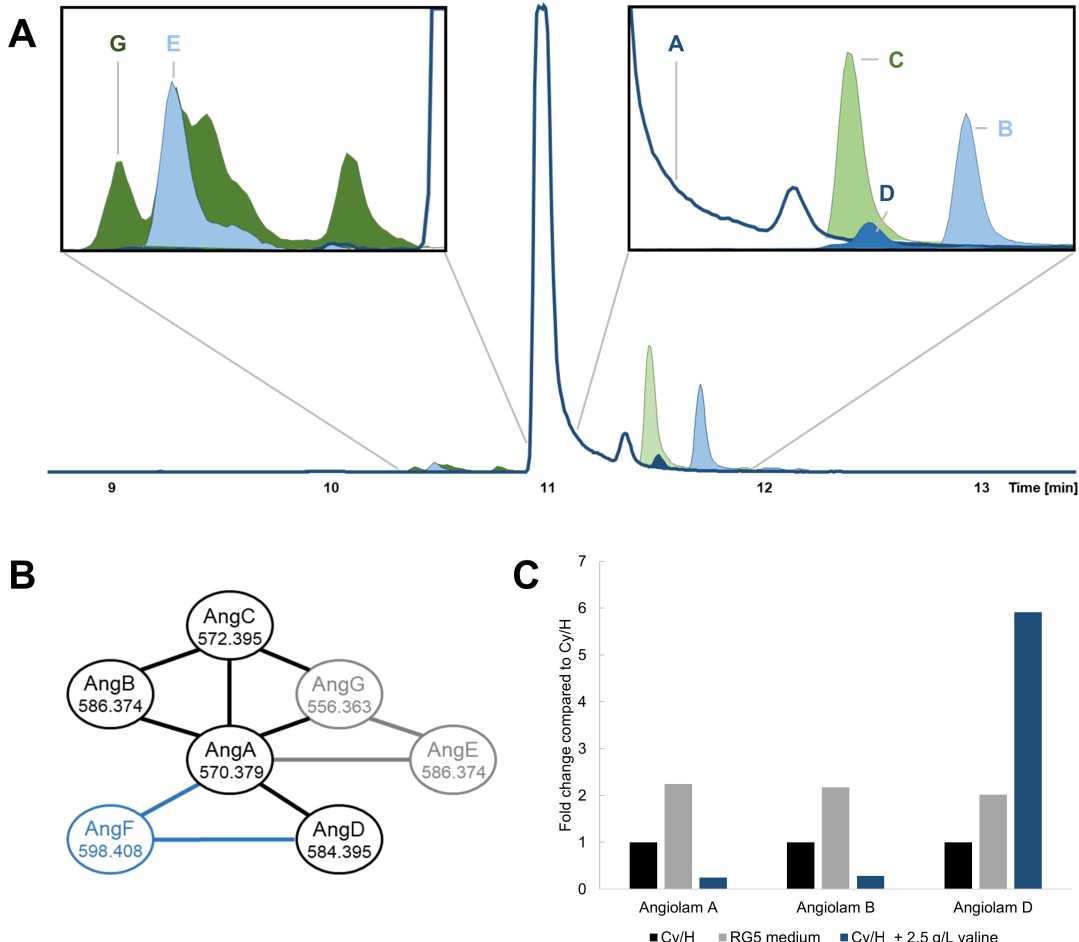

**FIG 2** The angiolam compound family and their production. (A) Extracted ion chromatograms of angiolam A (1, middle, dark blue/white) and its derivatives in MCy12733. Right box: angiolams B (2, light blue), C (3, light green), and D (4a/4b, blue). Left box: angiolams E (light blue) and G (dark green). (B) Feature-based molecular network, constructed by GNPS, of the angiolam compound family. Derivatives that were observed in Cy/H medium and subsequently purified are marked black. Angiolam F (5, blue) was observed and purified from Cy/H with additional 2.5 g/L L-valine. Angiolam derivatives that could not be purified due to low production yields are marked gray. The *m/z* ratios of the main ion in HPLC-MS measurements are indicated with the respective angiolam derivatives. Main ions, retention times, and sum formulae for the observed angiolam derivatives can be found in Table S9. (C) Relative production of angiolams A (1), B (2), and D (4a/4b) of MCy12733, when cultivated in different media. Data for all angiolam derivatives can be found in Tables S10 and S11.

subjected to an untargeted metabolome mining workflow, using feature detection with subsequent MS/MS-based molecular networking according to the GNPS method (28–30). Evaluation of these data showed that the MS/MS spectra of known angiolam A (1) cluster with MS/MS fragmentation spectra of five additional, to date unknown angiolam congeners (Fig. 2B).

As several of these observed angiolam-like compounds were produced in very low levels, a cultivation media screening was performed to improve their production. Cultivation of MCy12733 in RG5 medium improved the production of most observed angiolam derivatives two- to threefold (Fig. 2C; Tables S10 and S11). Analysis of the new extracts by MS/MS feature-based molecular networking showed the production of seven additional previously unknown angiolam derivatives, although mostly in trace amounts only (Fig. S1).

As production levels of the minor angiolams were small, the effect of added amino acids and polyketide precursors on the production of angiolam derivatives was tested. The addition of 2.5 g/L L-valine to Cy/H medium improved the production of novel angiolam D (4a/4b) about sixfold (Fig. 2C; Tables S10 and S11). Production of all other

angiolam derivatives was reduced about fourfold under these conditions making this a good example of precursor-guided steering of the angiolam biosynthesis. Further analysis of the extract with MS/MS feature-based molecular networking also led to the observation of novel angiolam F (5) that was only produced in cultivations with L-valine added to the production medium (Fig. 2B).

In order to allow compound isolation, angiolams were produced in large-scale cultivations of MCy12733 using different media and subsequently purified by preparative HPLC and, in the case of angiolam derivatives C (3) and D (4a/4b), further separated by supercritical fluid chromatography.

As part of compound purification efforts, angiolam A was also isolated and its known structure was verified. High-resolution electrospray-ionization mass spectrometry (HRESI-MS) of angiolam A (1) shows a $[M\text{-}H_2O + H]^+$ signal at $m/z$ 570.3785 (calc. 570.37889 $\Delta$ = 0.68 ppm) that corresponds to a molecular sum formula of $C_{34}H_{53}NO_7$ for [M]. The sum formula and the resulting nine double bond equivalents (DBEs) match the reported structure of angiolam A (24, 31). Comparison of $^1$H, $^{13}$C, and HSQC NMR spectra of 1 with authentic angiolam A confirmed the purified compound as angiolam A. Following the biosynthetic logic of angiolam A, we propose and employ a numeration for this family of natural products (Fig. 3).

Angiolam B (2) shows a HRESI-MS $[M + H]^+$ signal at $m/z$ 586.3748 (calc. 586.37385 $\Delta$ = 1.62 ppm), corresponding to the molecular sum formula of $C_{34}H_{51}NO_7$ that has an unsaturation degree of 10 DBEs. With respect to the respective spectra of angiolam A, the $^{13}$C NMR spectrum of 2 displays a further carbonyl signal at $\delta_{C\text{-}5}$ 210.2, and the HSQC NMR spectrum lacks the signal corresponding to the hydroxylated methine 5 in the side chain of 1. HMBC NMR correlations of the surrounding H-3, H-6a, H-6b, H-7a, H-7b, and H-28 to this carbonyl signal demonstrate its location as C-5.

The HRESI-MS signal of the angiolam C (3) ion $[M\text{-}H_2O + H]^+$ at $m/z$ 572.3955 (calc. 572.39455 $\Delta$ = 1.66 ppm) points to the molecular sum formula $C_{34}H_{55}NO_7$ for [M] with eight DBEs. Similar to angiolam B, angiolam C differs from angiolam A in the side chain. The HSQC NMR spectrum of 3 lacks the signals belonging to the terminal alkene of 1 but displays a new methyl group ($\delta_{C\text{-}1}$ 14.1 and $\delta_{H\text{-}1}$ 0.97) and a new methylene group ($\delta_{C\text{-}2}$ 20.8 and $\delta_{H\text{-}2}$ 2.03). COSY NMR correlations between this methylene group with the methyl group, as well as with H-3, show that angiolam C is fully saturated at the end of the side chain.

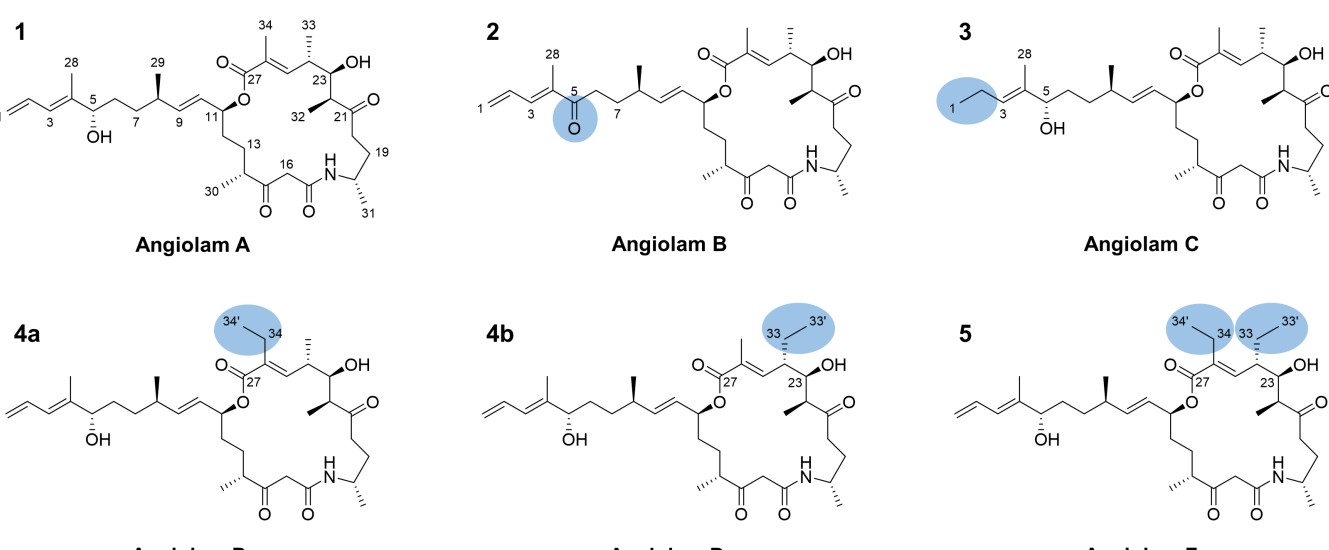

**FIG 3** Structures of known angiolam A and novel derivatives A, B, C, D₁, D₂, and F. Differences between angiolam A and novel derivatives are highlighted in the respective structures.

Evidence from the HRESI-MS [M-$H_2O$ + H]$^+$ signal at $m/z$ 584.3953 (calc. 584.39455 Δ = 1.28 ppm) suggested that angiolam D (4a/4b) possesses a molecular sum formula of $C_{35}H_{55}NO_7$ for [M] and nine DBEs. In contrast to the angiolam derivatives described above, angiolam D seems to be a mixture of one main and several minor components with very similar retention times (Fig. S2). Consequently, the purification workflow yielded the mixture of angiolam $D_1$ (4a) as major and angiolam $D_2$ (4b) as minor component. However, it was possible to elucidate the structures of 4a and 4b out of the same data set of 1D and 2D NMR spectra.

The difference between angiolams A and D is located within the macrocycle. While 1 holds methyl residues at C-24 and C-26, they are exchanged by an ethyl residue at C-26 in 4a and at C-24 in 4b. In 4a, this is demonstrated by the HSQC NMR signals of a methyl group ($\delta_{C-34'}$ 14.0 and $\delta_{H-34'}$ 1.01) and a diastereotopic methylene ($\delta_{C-34}$ 20.6, $\delta_{H-34a}$ 2.35, and $\delta_{H-34b}$ 2.29). The methyl protons H-34′ show COSY-NMR correlations to the methylene protons H-34a and H-34b and HMBC-NMR correlations to the olefinic C-26 ($\delta_{C-26}$ 135.1). Furthermore, the HMBC correlations of the methylene protons H-34a and H-34b to C-25 ($\delta_{C-25}$ 141.3), C-26 ($\delta_{C-26}$ 135.1), and C-27 ($\delta_{C-27}$ 167.4), as well as the HMBC correlation between H-25 ($\delta_{H-25}$ 6.39) and C-34, demonstrate the structure of angiolam $D_1$. The ethyl residue in 4b is indicated by the HSQC NMR signals of a diastereotopic methylene ($\delta_{C-33}$ 24.2, $\delta_{H-33a}$ 2.00, and $\delta_{H-33b}$ 1.33) and a methyl group ($\delta_{C-33}$ 11.2 and $\delta_{H-33}$ 0.86). COSY NMR correlations of methylene protons H-33a and H-33b to H-24 ($\delta_{H-24}$ 2.46), as well as to the methyl-group H-33′ ($\delta_{H-33'}$ 0.86), show that these signals belong to an ethyl residue that is bound to C-24. This is further evidenced by HMBC-NMR correlations of H-23 ($\delta_{H-23}$ 3.84) and H-25 ($\delta_{H-25}$ 6.41) to C-33 ($\delta_{C-33}$ 24.2) and reciprocal HMBC correlations between the methine H-24 ($\delta_{C-24}$ 43.2 and $\delta_{H-24}$ 2.46) and the methyl-group H-33′ ($\delta_{C-33}$ 11.2 and $\delta_{H-33}$ 0.86).

Angiolam F (5) shows a HRESI-MS [M-$H_2O$ + H]$^+$ signal at $m/z$ 598.4121 (calc. 598.41025 Δ = 3.09 ppm), corresponding to the molecular sum formula of $C_{36}H_{57}NO_7$ for [M] and nine DBEs. Similar to angiolam D, a mixture of one major and several minor components can be seen in the crude extracts (Fig. S2). However, the purification workflow resulted in one pure natural product. Compared to 1, the HSQC NMR spectrum of 5 lacks the methyl groups 33 and 34. Instead, it displays two slightly different methyl groups at $\delta_{C-33}$ 11.2, $\delta_{H-33}$ 0.86 and $\delta_{C-34}$ 13.6, $\delta_{H-34}$ 0.99 as well as two methylene groups at $\delta_{C-33}$ 24.2, $\delta_{H-33}$ 1.98 and $\delta_{C-34}$ 20.7, $\delta_{H-34}$ 2.34, similar to the signals belonging to the ethyl residues in 4a and 4b. As described above, COSY NMR correlations of methylene H-33 ($\delta_{H-33}$ 1.98) to H-24 ($\delta_{H-24}$ 2.45) and the methyl group H-33′ ($\delta_{H-33'}$ 0.86) identify an ethyl residue bound to C-24. Furthermore, methyl group H-34′ ($\delta_{H-34}$ 0.99) shows COSY correlations to methylene H-34 ($\delta_{H-34}$ 2.34) and HMBC-NMR correlations to olefinic C-26 ($\delta_{C-26}$ 136.6), pointing toward an ethyl residue, bound to C-26. The identity and location of both ethyl residues within the macrocycle are further evidenced by HMBC-NMR correlations, as elaborated for 4a and 4b.

Due to their low production yields, angiolam derivatives E and G could not be purified in amounts sufficient for NMR-based structure elucidation.

Considering a common biosynthetic route for angiolam A and its derivatives, it was deemed likely that they are produced by a set of enzymatic domains involved in polyketide synthase-nonribosomal peptide synthetase (PKS-NRPS) hybrid-type biosynthesis. In the biosynthetic assembly line, small parts of the PKS-NRPS megaenzymes, so-called domains control the incorporation of single building blocks into the growing scaffold and determine the stereochemistry of the emerging natural product. Therefore, we propose that the reported stereochemistry of angiolam A should also be found in all newly described derivatives (Fig. 3).

## *In silico* analysis of the angiolam BGC and biosynthesis hypothesis

The discovery and understanding of the biosynthetic pathway for a given natural product class provide valuable insights that can be used to improve the production

of these natural products or alter its biosynthesis towards the production of novel derivatives.

In order to find the biosynthetic origin of the angiolam compound family, genome sequences of both novel producers MCy12716 and MCy12733 were obtained by Illumina sequencing. Analysis of both sequences with antiSMASH (32) together with retro-biosynthetic considerations pinpointed a candidate polyketide synthase-nonribosomal peptide synthetase (PKS-NRPS) hybrid cluster with 12 modules spanning five genes. Alignment of the core BGC and its flanking regions was used to determine the likely cluster borders of the angiolam BGC. According to the similarities of both sequences, the *ang* BGC comprises a total of 14 genes spanning a 78.5 kb region. Detailed analysis of the genes in the *ang* BGC in MCy12733 was carried out by BLAST searches against the NCBI database (see below and Tables S20 and S21; Fig. S3). Interestingly, the genome of *P. caerfyrddinensis* CA032A, the closest described relative of MCy12733, harbors genes coding for proteins highly similar to the ones of the putative ang BGC. While these genes are present in the same order in both strains, *P. caerfyrddinensis* lacks the PKS-NRPS core as well as *angF*, encoding an acyl-CoA/acyl-ACP dehydrogenase. Therefore, it is somewhat likely the *ang* BGC only consists of the biosynthetic core *angA-angE* and *angF* or also includes the flanking genes *ang1-ang6* upstream and *ang7-ang8* downstream the core region that might also perform relevant functions for angiolam biosynthesis.

BLAST searches of *angA-angE* against draft genomes (Sebastian Walesch, unpublished data) of the known angiolam-producing strains *Pyxidicoccus fallax* An d30 (24, 25) and *P. fallax* An d48 (26) indicated the presence of the BGC but divided into several contigs due to poor DNA sequence quality. Furthermore, parts of the biosynthetic machinery, divided into at least two contigs, could be observed in the deposited genomes of *P. fallax* An d47 (GCA_012933655.1) and *P. fallax* CA059B (GCA_013155555.1).

In order to verify the compound-to-BGC assignment, the angiolam biosynthetic gene cluster was inactivated via single crossover plasmid insertion. As genetic manipulation of both new angiolam-producing strains MCy12716 and MCy12733 could not be achieved, the inactivation was performed in the genetically amenable producer *P. fallax* An d48 (26, 27) for which single crossover inactivation has already been shown to be achievable. A pCR2.1-TOPO-derived vector carrying a 1 kb homology fragment of the *angB* gene was integrated into the *angB* gene on the *P. fallax* An d48 chromosome. A mutant strain was selected on YM agar, supplemented with 50 µg/mL kanamycin. It was then cultivated for gDNA extraction and subsequent verification via PCR (Fig. S9), as well as to generate crude extracts to monitor angiolam production.

The single cross-over inactivation abolished the production of angiolam A (1, Fig. 4). Furthermore, no other angiolam derivative could be detected. This finding confirmed that *ang* BGC is the locus responsible for the production of the angiolam compound family.

Our biosynthetic findings agree with previous feeding experiments, revealing the biogenesis of angiolam A to comprise seven propionate equivalents, five acetate equivalents, and L-valine (24). According to predictions by antiSMASH (32) and domain fingerprints (33), the acyltransferases (ATs) of the loading module and modules 1, 3, 6, 10, 11, and 12 selectively incorporate methylmalonyl-CoA and the ATs of modules 2, 4, 5, 7, and 9 are selective for malonyl-CoA (Table S22; Fig. S4). The adenylation (A) domain of module 8 incorporates glycine or alanine, according to antiSMASH (Table S22). Analysis of the ketoreductase (KR), dehydratase (DH), and enoylreductase (ER) domains was done by antiSMASH, according to fingerprints (34, 35) or based on a hidden Markov model (36) (Table S23; Fig. S5 to S7). Deviations from the "colinearity rule" can be explained by inactive ER domains in modules 2 and 12, an inactive KR domain in module 10, a missing KR domain, as well as an inactive DH domain in module 7. Moreover, the *in silico* predictions of the stereochemistry based on KR, DH, and ER domains matched the actual stereochemistry of angiolam A.

An unusual feature of angiolam A is the distal alkene moiety, an altogether rare structural feature in natural products (37). The terminal alkenes in curacin A and

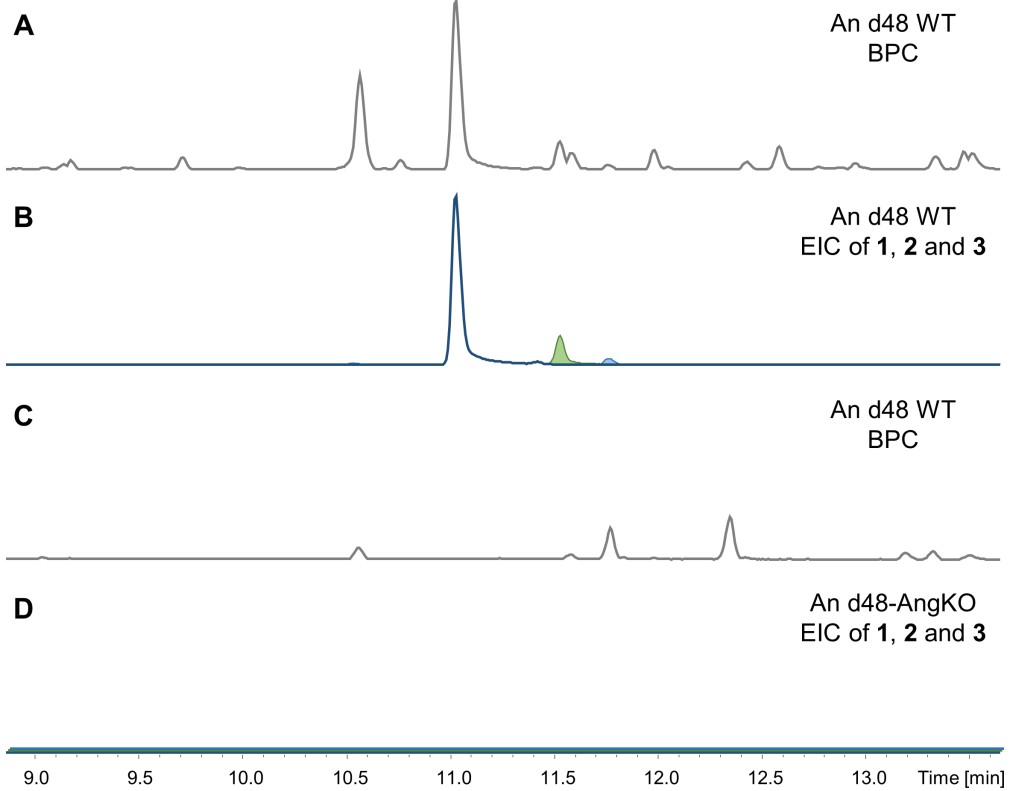

**FIG 4** Analytical verification of the angiolam BGC assignment. UHPLC MS extracted ion chromatogramEIC traces of angiolams A, B, and C (B, D; blue, red, and green) as $[M-H_2O + H]^+$ ions (1, 570.379 Da and 3, 572.395 Da) and as $[M + H]^+$ ions (2, 586.374 Da) in the wild type (A and B) and in the angB single cross-over inactivation mutant (C and D) crude extracts. Gray traces represent the corresponding base peak chromatograms (BPC). Intensity scale in D is magnified 100-fold. EIC traces in panel D are stacked for a better overview.

tautomycetin are formed while or after they are released from the respective megasynthase (38, 39). Similar to angiolam A, the alkene starter in haliangicin A is part of its starter unit and is formed by the acyl-CoA dehydrogenase HliR through a γ,δ-dehydrogenation of 2-methylpent-2-enoyl-CoA (40). Based on this finding, a possible explanation for the distal alkene moiety of angiolam A could be a dehydrogenation of the methylmalonyl-CoA-derived starter unit to prop-2-enoyl-ACP by the acyl-CoA/acyl-ACP dehydrogenase AngF in the loading module (Fig. 5B). However, dehydrogenation to form the distal alkene is also possible later, during or after the assembly line biosynthesis.

Following the incorporation of the starter unit, angiolam A undergoes seven PKS-derived elongation steps including the incorporation of four malonyl-CoA and three methylmalonyl-CoA extender units with varying oxidation states, based on the respective reductive loops. Next, L-alanine is incorporated by the NRPS module 8. This is followed by four further PKS-derived elongation steps of one malonyl-CoA and three methylmalonyl-CoA extender units to form the angiolam A backbone. Finally, the fully elongated intermediate is cyclized and released from the PKS-NRPS machinery by the thioesterase domain in module 12 (Fig. 5A).

The likely explanations for the structural differences observed in angiolams B and C are incomplete reduction by the KR domain in module 2 and dehydrogenation reactions at the end of the angiolam side chain, respectively. The observation of the ethyl residues in modules 11 and 12 in angiolams D and F is more surprising. As feeding experiments with L-methionine-(methyl $^{13}$C) did not result in mass shifts (Fig. S8), a SAM-dependent methylation of the respective methyl residues seemed unlikely. Therefore, we concluded that these ethyl residues originate from ethylmalonyl-CoA extender units and are incorporated by AT domains with broader substrate specificities. A prominent example

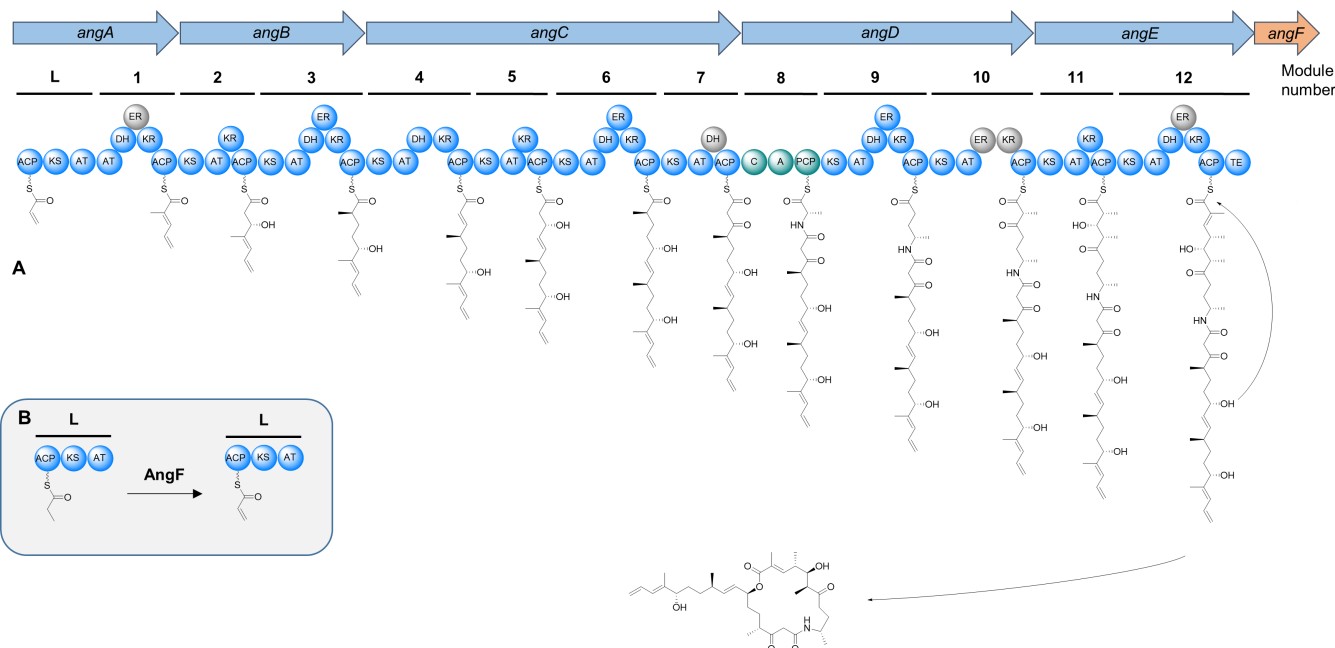

**FIG 5** *In silico* biosynthesis of angiolam A. (A) Scheme of the proposed angiolam BGC and the angiolam biosynthesis model by its megasynthase. (B) Potential biosynthesis of the prop-2-enoyl-ACP starter unit via dehydrogenation of propanoyl-ACP by AngF (arrows, genes; circles, domains; A, adenylation; ACP/PCP, acyl-/peptidyl-carrier protein; AT, acyl transferase; C, condensation; DH, dehydratase; ER, enoylreductase; KR, ketoreductase; KS, ketosynthase; and TE, thioesterase).

of an AT domain that incorporates methylmalonyl-CoA and ethylmalonyl-CoA is the AT domain of module 5 of the monensin biosynthesic gene cluster (41). Screening of the ang AT domains for the reported signature sequence RiDVV (41) revealed that not only modules 11 and 12 but also modules 3, 6, and 10 might have the capacity to accept ethylmalonyl-CoA as an extender unit (Fig. S4), leading to the observed natural structural diversity of angiolams. We reason that this also explains the finding of angiolams D and F as mixtures of several components. However, the reasons for the increased incorporation of ethylmalonyl-CoA by modules 11 and 12 in comparison to the others remain elusive. Interestingly, AT domains in modules 2, 5, 6, 7, and 8 in the epothilone BGC in *Sorangium cellulosum* So ce90 (42) also display the signature sequence RiDVV. So far, there is only one report about an epothilone derivative with an ethyl residue, putatively incorporated by one of these AT domains (43), indicating that this signature sequence might point toward AT domains that accept either methyl- or ethylmalonyl CoA as their substrate. However, the prioritization of the substrate seems to depend on other parameters like other sequence motifs within the AT domains or its three-dimensional structure, KS domain substrate specificities, or precursor supply.

The importance of precursor supply for the incorporation of ethylmalonyl-CoA is demonstrated by the increased production of angiolams D and F in cultivations with added ʟ-valine, as it serves as a precursor for ethylmalonyl-CoA (44). As ʟ-valine can also be a precursor of methylmalonyl-CoA (44), feeding experiments with ʟ-valine $^{13}C_5$ resulted in inconclusive results (data not shown). However, the role of ʟ-valine as a precursor for the ethylmalonyl-CoA supply in MCy12733 is further evidenced by the presence of a gene encoding a fused isobutyryl-CoA mutase located elsewhere within its genome (Tables S19 and S20). Furthermore, the addition of butyrate, a possible intermediate in the pathway from ʟ-valine to ethylmalonyl-CoA, also increased the production of angiolams D and F (Table S24) reinforcing this hypothesis.

## Antiparasitic activities

Based on the antiparasitic bioactivity of the crude extract of MCy12716, angiolams A (1), B (2), C (3), D (4), and F (5) were tested for their *in vitro* activities against the following protozoan parasites: *Trypanosoma brucei rhodesiense* (STIB 900) bloodstream forms, *Trypanosoma cruzi* (Tulahuen C4) amastigotes, *Leishmania donovani* (MHOM/ET/67/L82) axenic amastigotes, and *Plasmodium falciparum* (NF54) proliferative erythrocytic stages. In parallel, the cytotoxicity of these compounds in rat skeletal myoblasts (L-6 cells) was determined in order to obtain an initial assessment of their selectivity. The results are reported in Table 1.

### Activity against trypanosomatids

Comparing the activity of different angiolam congeners against *Trypanosoma brucei rhodesiense*, there is a clear matching between the antitrypanosomal activity and selectivity (A > C > D1 >B > F). Angiolam A showed the highest activity (IC$_{50}$ of 1.5 µM) toward *T. brucei* as well as preferential selectivity across all the three trypansomatids. Its trypanocidal activity is comparable to that previously described for Macyranone A, a peptide-epoxyketone isolated from another myxobacterial strain *Cystobacter fuscus* MCy9118 (11). Angiolams B, C, and D showed similar activity and toxicity profiles against *T. b. rhodesiense* (IC$_{50}$ range: 4.5–6.3 µM, SI: 4.4–6.8), while angiolam F was the least active and most cytotoxic.

Trypanosoma cruzi was the least sensitive parasite (IC$_{50}$: 6.6–34.0 µM) with unfavorable selectivities displayed by most of the tested compounds (angiolams B, C, and F) due to their relatively high cytotoxicity (SI ≤ 2.1). Although both *T. brucei* and *T. cruzi* are trypansomatids, however, the lack of overlapping activity is not unusual (45). This could be partially explained by the intracellular nature of the parasite (46). Nevertheless, angiolam D was found to be the most active (IC$_{50}$: 6.5 µM), with twofold higher activity with respect to angiolam A. This could be attributed to the CH$_3$ extension at C-33 or C-34, which, on the other hand, diminished the selectivity.

Angiolams exhibited moderate to low activity (IC$_{50}$: 6.6–17.6 µM) and unsatisfactory selectivity (selectivity indices of 2–6) toward *L. donovani* axenic amastigotes. Angiolam B has shown the highest activity (IC$_{50}$: 6.6 µM), though 20-fold weaker than its displayed activity against *P. falciparum*, followed by angiolam D (IC$_{50}$: 8.2 µM). Angiolams A, C, and F showed comparable activities (IC$_{50}$: 15.4–17.6 µM), with angiolam C being the most cytotoxic.

### Activity against Plasmodium falciparum

The *in vitro* activity of angiolams against the erythrocytic stages of the *P. falciparum* strain NF54 was the most promising compared to the other parasites tested. Angiolams

TABLE 1  *In vitro* activity of compounds 1–5 against *L. donovani* (MHOM-ET-67/L82) axenic amastigotes, *P. falciparum* (NF54), *T. b. rhodesiense* (STIB 900), *T. cruzi* (Tulahuen C4), and cytotoxicity in L6 cells

| ID No | T. b. rhodesiense | | T. cruzi | | L. donovani | | P. falciparum | | Cytotoxicity in L6 cells |
|---|---|---|---|---|---|---|---|---|---|
| | IC$_{50}$$^a$ (µM) | SI$^b$ | IC$_{50}$$^a$ (µM) | SI$^b$ | IC$_{50}$$^a$ (µM) | SI$^b$ | IC$_{50}$$^a$ (µM) | SI$^b$ | IC$_{50}$$^a$ (µM) |
| Angiolam A (1) | 1.5 ± 0.6 | 63.2 | 12.9 ± 5.3 | 7.3 | 16.7 ± 7.3 | 5.62 | 2.7 ± 0.5 | 34.9 | 94.0 ± 23.7 |
| Angiolam B (2) | 6.3 ± 1.0 | 4.4 | 30.4 ± 10.0 | 0.9 | 6.6 ± 1.9 | 4.22 | 0.3 ± 0.1 | 91.6 | 27.8 ± 2.0 |
| Angiolam C (3) | 4.5 ± 0.4 | 6.8 | 34.0 ± 11.6 | 0.9 | 15.4 ± 2.3 | 1.99 | 0.6 ± 0.4 | 53.7 | 30.6 ± 2.3 |
| Angiolam D (4) | 4.7 ± 1.9 | 6.0 | 6.5 ± 2.2 | 4.3 | 8.2 ± 4.5 | 3.38 | 0.8 ± 0.6 | 37.1 | 27.8 ± 3.7 |
| Angiolam F (5) | 25.0 ± 1.4 | 2.0 | 24.4 ± 5.0 | 2.1 | 17.6 ± 2.1 | 2.83 | 1.3 ± 0.6 | 37.4 | 49.9 ± 21.6 |
| Positive control | 0.02$^c$ | | 2.47$^d$ | | 0.53$^e$ | | 0.004$^f$ | | 0.02$^g$ |

$^a$The IC$_{50}$s are mean values from at least three independent replicates ± standard deviation.
$^b$Selectivity index (SI): IC$_{50}$ in L6 cells divided by IC$_{50}$ in the titled parasitic strain.
$^c$Melarsoprol.
$^d$Benznidazole.
$^e$Miltefosine.
$^f$Chloroquine.
$^g$Podophyllotoxin.

B, C, and D showed activity in submicromolar concentrations (IC$_{50}$: 0.3–0.8 µM) with high selectivities (SI ≥ 30). The most promising compound of this series was angiolam B exhibiting submicromolar activity (IC$_{50}$: 0.3 µM) and highest selectivity (SI: 91.6). We assume that this could be due to the oxidation of C-5 that has an augmenting effect on both activity and selectivity. Angiolams D and F showed IC$_{50}$ values of 0.8 and 1.3 µM, respectively. The presence of two ethyl groups in angiolam F may have contributed to the reduced activity; however, this substitution did not show an effect on the selectivity (SI 37 for both D and F). Unlike its antitrypanosomal activity, angiolam A showed the least antiplasmodial activity and selectivity among the tested angiolam derivatives.

## Conclusion and outlook

This study demonstrates the prospects of a regional sampling campaign to bring forward new myxobacterial natural product producers, in combination with screening efforts that go beyond antibacterial or cytotoxic activities. The uncovering of new producing strains with unexpected bioactivities reinvigorated the interest in angiolam almost 40 years after it was first reported as an antibiotic (24, 25). This interest subsequently led to the characterization of several novel angiolam derivatives with improved bioactivities against *P. falciparum*. It also initiated the elucidation of the underlying biosynthesis, including a potentially rare enzymatic dehydrogenation of the starter unit. Moreover, our present data highlight the potential of the angiolam scaffold as a starting point for the development of novel drugs with anti-parasitic bioactivities. This view is supported by the promising selectivity index of most angiolam derivatives against *P. falciparum* in combination with the reported tolerability of angiolam A in mice as well as the availability of a total synthesis route (25, 31).

Taken together, this study illustrates the opportunities of reinvestigating "old" natural products with improved analytical and computational resources at hand. Given the energetic effort that microorganisms put into the biosynthesis of these compounds, it is likely that the competitive advantage conveyed by these natural products comes with additional biological activities awaiting their future discovery through suitable bioassays.

## MATERIALS AND METHODS

### Isolation of myxobacteria

Both myxobacterial strains MCy12716 and MCy12733 originated from the citizen-science campaign "Sample das Saarland," a regional project initiated in 2017 that aims to discover novel antibiotics and new anti-infectives from local samples. The first strain MCy12716 was isolated in March 2019 based on the standard bacterial predation method using *Escherichia coli*. This strain came from pond soil sediments collected in the summer of 2018 (03 June) in an area between Karlsbrunn and Ludweiler, Germany (coordinates: 49.20153847434484,6.800456559291888). On the other hand, the second myxobacterium MCy12733 was isolated after a month (April 2019) but from a soil sample with decaying leaves collected on 09 August 2018 at Aßweiler, Germany (GPS coordinates: 49.2189122,7.192166499999985). Its discovery was also based on *E. coli* lysis, formation of swarming colony, and development of fruiting bodies on Stan-21 mineral salt agar medium. Axenic cultures were achieved by repeated transferring of the swarm cells taken from the colony margin and by maintaining in VY/2 agar, which supports growth developmental stages, including swarming and fruiting body formation. These two myxobacterial isolates were identified by 16S rRNA gene amplification using universal 27F and 1492R primers. Their closest related type strains were determined by BLAST search and phylogenetic analysis.

### Fermentation of myxobacteria for analytical purposes

Cultures for analytical purposes were grown in 300 mL shake flasks containing 50 or 100 mL of the respective fermentation medium inoculated with 2% (vol/vol) pre-culture.

The medium was supplemented with 4% (vol/vol) of a sterile aqueous solution of XAD-16 adsorber resin (Sigma Aldrich) to bind secondary metabolites in the culture medium. After 7–10 days of cultivation, the culture is pelleted in 50 mL falcon tubes in an Eppendorf centrifuge 5804R at 8,288 × g and 4°C for 10 min. The pellet is then stored at −20°C until further use.

## Feeding experiments with stable isotope-labeled precursors

Feeding experiments were performed in 100 mL shake flasks containing 20 mL of the respective fermentation medium inoculated with 2.5% (vol/vol) SBCy413 pre-culture. At 12-, 24-, 36-, 48-, and 60-h post-inoculation, 200 µL of an aqueous solution of 50 mM L-alanine $^{15}$N (Sigma Aldrich) or L-methionine-(methyl $^{13}$C) (Cambridge Isotope Laboratories) was added to the cultures. After another 12 h of cultivation, 5% (vol/vol) sterile aqueous solution of XAD-16 adsorber resin was added to the cultures. After approximately 4 days of cultivation, the culture is pelleted in 50 mL falcon tubes in an Eppendorf centrifuge 5804R at 8,288 × g and 4°C for 10 min. The pellet is then stored at −20°C until further use.

## Large-scale fermentation for compound purification

Pre-cultures of MCy12733 were cultivated in 300 mL shake flasks containing 100 mL of the respective fermentation medium at 30°C and 180 rpm for 5 days. These pre-cultures are used to inoculate [5% (vol/vol) inoculum] 6 × 2 L of the respective fermentation medium, supplemented with 5% (vol/vol) of a sterile aqueous solution of XAD-16 adsorber resin in 5 L shaking flasks. Fermentation was performed at 30°C and 170 rpm for 21 days. Afterward, the combined cultures were pelleted with an Avanti J-26 XP centrifuge, equipped with a JLA-8.1 rotor (Beckman Coulter) at 11,978 × g and 4°C for 15 min. The pellet was then frozen, freeze-dried, and stored at −20°C until further use.

## Preparation of gDNA

Genomic DNA of *P. fallax* An d48 for cloning purposes and *P. fallax* An d48-AngKO to verify the integration of pTOPO-AngKO within angB was isolated using the Puregene Core Kit A (Qiagen) or the DNeasy UltraClean Microbial Kit (Qiagen), according to the manufacturer's instructions.

To isolate the gDNA of MCy12716 and MCy12733 for Illumina sequencing, standard phenol-chloroform extraction was used.

## Whole-genome sequencing

The genome sequence of strain MCy12733 was acquired using Illumina technology to give 88 assembled contigs with a total sequence length of 13.77 Mbp.

## PCR amplification

Amplification of a genetic region in *angB* for gene disruption by single cross-over and PCRs to verify the integration of the plasmid into the angiolam BGC in *P. fallax* An d48 were done with the Phusion High-Fidelity polymerase (Thermo Scientific). All PCR amplifications were performed in a MasterCycler Pro S (Eppendorf). The success of PCR amplifications was checked by agarose gel electrophoresis. Bands of PCR products for cloning purposes were cut out and purified using the Nucleospin Gel and PCR Clean-Up Kit (Macherey-Nagel). Further information about PCR mixes and cycler protocols can be found in the supplemental material (Tables S5 to S8).

## Restriction enzyme digestion protocols

Restriction enzyme digestions for cloning purposes were performed in volumes of 20 µL with 350 ng PCR product or 1.2 µg cyclized pCR2.1-TOPO vector, respectively. Aqueous solutions of the PCR product or the cyclized pCR2.1-TOPO vector were mixed with 2 µL

of 10× EcoRI buffer and 1 µL of EcoRI (10 U/µL, Thermo Scientific) and incubated at 37°C. After 2 h, 1 µL of FastAP (1 U/µL, Thermo Scientific) was added to the reaction mixture of the vector to dephosphorylate its ends. All digestion reactions were stopped after 3 h of incubation, and the resulting products were subsequently purified, using the Nucleospin Gel and PCR Clean-Up Kit (Macherey-Nagel).

Control digestions to verify the success of cloning were performed in volumes of 10 µL with approximately 1 µg plasmid. Aqueous solutions of cloned plasmid were mixed with 1 µL of Tango buffer (Thermo Scientific) and 0.5 µL of NcoI (10 U/ µL, Thermo Scientific) or 0.5 µL of NaeI (10 U/ µL, Thermo Scientific) and incubated at 37°C for 2 h. Successful cloning was verified by agarose gel electrophoresis and sequencing of the obtained plasmid.

## Ligation protocol

Ligation of DNA was carried out with ~50 ng vector and a fivefold molar excess of insert. Vector and insert were mixed with 2 µL of 10× T4 buffer, 1 µL T4 ligase (5 U/ µL; Thermo Fisher Scientific), and autoclaved Milli-Q water to a final volume of 20 µL. Reactions were incubated at 25°C for 4 h or at 16°C overnight and subsequently electroporated into *E. coli* HS996.

## Transformation of *E. coli* HS996

One milliliter of an overnight culture of *E. coli* cells in LB was inoculated into 20 mL of LB medium and grown at 37°C until it reached an $OD_{600}$ of 0.6. The cells were centrifuged at $11,467 \times g$ and 4°C (HIMAC CT15RE, Koki Holdings Co.) for 90 s in 2 mL Eppendorf tubes. The cell pellet was washed twice with ice-cold sterile MQ water, first with 1,000 µL, then with 800 µL. After the washing procedure, *E. coli* cells were resuspended in 50 µL ice-cold sterile MQ water, kept on ice, and subsequently used for electroporation.

*E. coli* cells were mixed with 3–5 µL ligation product or 100 ng plasmid and transferred to an electroporation cuvette. Electroporation was carried out at 25 µF, 200 Ω, and 1,250 V. The electroporation product was immediately mixed with 1 mL LB medium and incubated in 2 mL Eppendorf tubes at 37°C and 750 rpm for 45 min. Incubated cells were spread on LB-agar plates containing 50 µg/mL kanamycin and incubated overnight at 37°C. Around 10 clones were picked and cultivated in 5 mL LB containing 50 µg/mL kanamycin for subsequent plasmid isolation.

## Plasmid isolation

Plasmid isolation for cloning purposes or transformation into *P. fallax* An d48 was performed with the GeneJET Miniprep Kit (Thermo Scientific). Plasmid isolation for control digestions to verify successful cloning was done with standard alkaline lysis (47).

## Transformation into *P. fallax* An d48

Electroporation of *P. fallax* An d48 was adapted from Panter et al. (27). Two milliliters of an An d48 culture in YM medium at $OD_{600}$ of approximately 0.7 were centrifuged at $11,467 \times g$ (HIMAC CT15RE, Koki Holdings Co.) for 2 min. The pelleted cells were washed twice with sterile MQ water, first with 1,000 µL, then with 800 µL. After the washing procedure, the cells were resuspended in 50 µL sterile MQ water, mixed with 5 µL of plasmid solution at a concentration of 0.3–0.4 µg/µL, and moved to an electroporation cuvette. Electroporation was performed at 675 V, 400 Ω, 25 µF, and 1 mm cuvette width. Afterward, the cells were flushed out with 1 mL YM medium and moved to a 2 mL Eppendorf tube with a small hole in the lid. After incubation at 30°C and 1,000 rpm for 5 h, the cell suspension was mixed with 3 mL YM soft agar, containing 50 µg/mL kanamycin, and the mixture was distributed on a YM agar plate containing 50 µg/mL kanamycin. Slightly red, spherical transformants could be observed after 10–14 days of cultivation at 30°C.

## Crude extract generation for analytical purposes

Frozen or freeze-dried pellets from small-scale cultivations were suspended in 40 or 80 mL methanol and stirred at 250 rpm at room temperature for 2 h. The supernatant was decanted into a round flask through a 125-µm folded filter. The solvent and potential residual water were removed on a rotary evaporator with a water bath temperature of 40°C at appropriate pressures. The dried extract was dissolved/resuspended in 1,100 µL MeOH/100 mL cultivation volume. The crude extract was stored at −20°C until further analysis. For the purpose of UHPLC-hrMS analysis, the crude extract was diluted 1:5 with methanol and centrifuged at 21,500 × $g$ and 4°C (HIMAC CT15RE, Koki Holdings Co.) for 5 min to remove residual insolubilities such as salts, cell debris, and XAD fragments.

## Standardized UHPLC-MS conditions

UPLC-hrMS analysis was performed on a Dionex (Germering, Germany) Ultimate 3000 RSLC system using a Waters (Eschborn, Germany) BEH C18 column (50 × 2.1 mm, 1.7 µm) equipped with a Waters VanGuard BEH C18 1.7 µm guard column. Separation of 1 µL sample was achieved by a linear gradient from (A) $H_2O$ + 0.1% formic acid (FA) to (B) ACN + 0.1% FA at a flow rate of 600 µL/min and a column temperature of 45°C. Gradient conditions for crude extract analysis were as follows: 0–0.5 min, 5% B; 0.5–18.5 min, 5%–95% B; 18.5–20.5 min, 95% B; 20.5–21 min, 95%–5% B; and 21–22.5 min, 5% B. Following gradient conditions were applied to monitor angiolams during purification: 0–0.5 min, 5% B; 0.5–9.5 min, 5%–95% B; 9.5–10.5 min, 95% B; 10.5–11 min, 95%–5% B; and 11–12.5 min, 5% B. UV spectra were recorded by a DAD in the range from 200 to 600 nm. The LC flow was split to 75 µL/min before entering the Bruker Daltonics maXis 4G hrToF mass spectrometer (Bremen, Germany) equipped with an Apollo II ESI source. Mass spectra were acquired in centroid mode ranging from 150 to 2,500 $m/z$ at a 2 Hz full scan rate. Mass spectrometry source parameters were set to 500 V as end plate offset; 4,000 V as capillary voltage; nebulizer gas pressure 1 bar; dry gas flow of 5 L/min; and a dry temperature of 200°C. Ion transfer and quadrupole settings were set to funnel RF 350 Vpp; multipole RF 400 Vpp as transfer settings and ion energy of 5 eV as well as a low mass cut of 300 $m/z$. The collision cell was set to 5.0 eV and pre-pulse storage time was set to 5 µs. The spectra acquisition rate was set to 2 Hz. Calibration was done automatically before every LC-MS run by injection of a sodium formate solution and calibration on the respective sodium formate clusters formed in the ESI source. All MS analyses were acquired in the presence of the lock masses ($C_{12}H_{19}F_{12}N_3O_6P_3$, $C_{18}H_{19}O_6N_3P_3F_2$, and $C_{24}H_{19}F_{36}N_3O_6P_3$), which generate the [M + H]$^+$ ions of 622.0289, 922.0098, and 1221.9906.

## Statistics-based metabolome filtering

For statistical metabolomics analysis, both the myxobacterial culture and medium blanks were incubated and extracted in triplicates as described above. Each extract was measured as a technical duplicate, giving a total number of six replicates each for the bacterial culture and medium blank extracts. T-ReX-3D molecular feature finder of MetaboScape 2022b (Bruker Daltonics, Billerica, MA, USA) was used to obtain molecular features. Detection parameters were set to an intensity threshold of 5 × 10$^3$ and a minimum peak length of five spectra to ensure a precursor selection that has a sufficient intensity to generate tandem MS data. Identification of bacterial features was performed with the built-in ANOVA/$t$-test routine and filtered to appearance in all six bacterial extracts and in none of the medium blank extracts. The $t$-test table was used to create a scheduled precursor list (SPL) for UHPLC-tandem MS analysis (28).

## Acquisition parameters for high-resolution tandem MS data

LC and MS conditions for SPL-guided MS/MS data acquisitions were kept constant according to section standardized UHPLC-hrMS conditions. MS/MS data acquisition

parameters were set to exclusively fragment scheduled precursor list entries. SPL tolerance parameters for precursor ion selection were set to 0.2 min and 0.05 $m/z$ in the SPL MS/MS method. The method picked up to two precursors per cycle, applied smart exclusion after five spectra, and performed CID and MS/MS spectra acquisition time ramping. CID Energy was ramped from 35 eV for 500 $m/z$ to 45 eV for 1,000 $m/z$ and 60 eV for 2,000 $m/z$. MS full scan acquisition rate was set to 2 Hz, and MS/MS spectra acquisition rates were ramped from 1 to 4 Hz for precursor ion intensities of 10–1,000 kcts.

## Parameters for feature-based molecular networking

All supporting feature-based molecular networking data presented here were created based on the UHPLC-hrMS² chromatograms using the parameters specified in the previous section. All molecular features were obtained by Metaboscape 2022b, with the same parameters as in Statistics-based metabolome filtering, and the respective MS/MS spectra were detected by the built-in "extract MS/MS data" function. The resulting output files were exported with the built-in "Export for GNPS/Sirius" function and uploaded to the GNPS server at the University of California San Diego via FileZilla FTP upload to ftp:// ccms-ftp01.ucsd.edu and all acquired SPL MS/MS spectra were used for feature-based network creation (29). A molecular network was created using the online workflow at GNPS. The data were filtered by removing all MS/MS peaks within ±17 Da of the precursor $m/z$. MS/MS spectra were window filtered by choosing only the top six peaks in the ±50 Da window throughout the spectrum. The data were then clustered with a parent mass tolerance of 0.02 Da and an MS/MS fragment ion tolerance of 0.02 Da to create consensus spectra. No further filtering of consensus spectra was done before spectral network creation. A network was then created where edges are filtered to have a cosine score above 0.7 and more than six matched peaks. Further edges between two nodes were kept in the network if and only if each of the nodes appeared in each other's respective top 10 most similar nodes (29). Furthermore, library search parameters were set to a cosine of 0.7 and a minimum of six matched peaks. Library searches included analogs with mass differences up to 200 Da. The data set was downloaded from the server and subsequently visualized using Cytoscape 3.9.1.

## Extraction and liquid/liquid partitioning

Extraction and liquid/liquid partitioning of cultures of MCy12733 in different production media were done with 100 mL of the respective solvents per 10 g freeze-dried pellet.

Freeze-dried pellets were extracted four times with a 1:1 mixture of methanol and acetone. For each extraction step, the suspension was stirred at 250 rpm for 30 min and decanted through glass wool and a 125-μm folded filter. The combined extracts were dried using a rotary evaporator at 40°C water bath temperature and appropriate pressures. The dried extract was then dissolved/resuspended in a 95:5 mixture of methanol and water and defatted three times with hexane. Finally, the defatted extract was partitioned three times between water and chloroform. The angiolam derivatives were retained in the chloroform phase. The chloroform phase was concentrated, moved to glass vials, dried, and stored at −20°C.

## Pre-purification by reverse phase flash chromatography

Reverse phase flash chromatography was carried out using the Isolera One (Biotage). Versa Flash Spherical C18 silica 45–75 μm, 70 Å in a 25 g SNAP column (Biotage) was used as the stationary phase, and $H_2O$ (A) and MeOH (B) were used as the mobile phase. Flow rate was 25 mL/min, fractions of 45 mL were collected in glass tubes and solvent volume was measured in column volumes (CV); 1 CV = 33 mL. After equilibration of the system with 3 CV of 50% B, ~300 μg sample was loaded on the column using vacuum-dried iSolute beads (Biotage). The conditions were kept at 50% B for 2 CV, followed by an increase to 80% B over 30 CV, a ramp to 100% B over 5 CV, and a flushing step at 100%

B for 5 CV. Fractions were pooled according to the abundance of angiolam derivatives, dried, and stored in glass vials at −20°C.

## Purification of angiolams A, B, and F

Purification of angiolams A (1), B (2), and F (5), as well as a mixture of angiolam C (3) and D (4a/4b) from the respective pooled flash chromatography fractions, was carried out on the Waters Autopurifier (Eschborn, Germany) high-pressure gradient system. Separation was carried out on a Waters X-Bridge prep C-18 5 µm, 150 × 19 mm column using $H_2O$ + 0.1% FA (A) and ACN + 0.1% FA (B) as mobile phase at a flow rate of 25 mL/min. Separation was started with 5% B for 1 min, followed by a ramp to 40% B over 2 min and an increase to 70% B over 22 min. The column was then flushed with a ramp to 95% B over 2 min, brought back to 5% B within 1 min, and re-equilibrated at 5% for 2 min. Purified angiolams A (1), B (2), and F (5) were dried with a rotary evaporator, moved to glass vials, freeze-dried, and stored at −20°C.

## Purification of angiolams C and D

The mixture of angiolams (3) and D (4a/4b) was separated on a Waters Prep 15 SFC System equipped with a 5 µm Torus Diol 130 Å OBD Prep Column 250 × 19 mm (Waters) thermostated at 50°C. Separation was achieved by following multistep gradient of methanol as a co-solvent to the supercritical $CO_2$. A ramp of 5%–11% co-solvent within 1 min was followed by an increase to 17% over 18 min to separate angiolams (3) and D (4a/4b). Afterward, the column was flushed with an increase of 20% co-solvent for 1 min, the co-solvent was decreased to 5% within 0.5 min, and the system was re-equilibrated at 5% co-solvent for 2.5 min. The flow rate was set to 15 mL/min, make-up flow to 3 mL/min, and backpressure to 120 bar.

## NMR-based structure elucidation

The chemical structures of angiolam derivatives were determined via multidimensional NMR analysis. 1D and 2D NMR spectra were recorded at 500 MHz ($^1$H) and 125 MHz ($^{13}$C) or 700 MHz ($^1$H) and 175 MHz ($^{13}$C) conduction using an Avance III (Ultrashield) 500 MHz spectrometer or an Avance III (Ascend) 700 MHz spectrometer, respectively, both equipped with a cryogenically cooled triple resonance probe (Bruker Biospin Rheinstetten, Germany). All observed chemical shift values ($\delta$) are given in ppm and coupling constant values ($J$) in Hz. Standard pulse programs were used for HMBC, HSQC, and gCOSY experiments. HMBC experiments were optimized for $^{2,3}J_{C\text{-}H}$ = 6 Hz. The spectra were recorded in chloroform-$d$, and chemical shifts of the solvent signals at $\delta_H$ 7.27 ppm and $\delta_C$ 77.0 ppm were used as reference signals for spectra calibration. To increase sensitivity, the measurements were conducted in a 5 mm Shigemi tube (Shigemi Inc., Allison Park, PA, USA). The NMR signals are grouped in the tables in the supplemental material and correspond to the numbering in the schemes corresponding to every table.

## *In vitro* antiparasitic activity assays

*In vitro* activity assays against a panel of protozoal parasites, namely *Trypanosoma brucei rhodesiense* (STIB 900), *Trypanosoma cruzi* (Tulahuen C4), *Leishmania donovani* (MHOM/ET/67/L82), and *Plasmodium falciparum* (NF54), and cytotoxicity tests against mammalian cells (L6-cell line from rat-skeletal myoblasts) were performed according to previously described methods (2, 3). All assays were performed in at least three independent replicates.

## ACKNOWLEDGMENTS

The authors thank Joy Birkelbach for the verification and proofreading of the structure elucidation and scientific discussions, Nestor Zaburannyi for genome assembly, and Clemens Schumm for his assistance in the verification of the gene inactivation.

Furthermore, we wish to thank Monica Cal, Romina Rocchetti, and Sonja Keller-Märki for their assistance with the parasite and cytotoxicity assays.

Research in R.M.'s laboratory is funded by the Helmholtz Association (HGF), the German Research Foundation (DFG), the Federal Ministry of Education and Research, Germany (BMBF), and the German Center for Infection Research (DZIF).

## AUTHOR AFFILIATIONS

[1]Helmholtz Centre for Infection Research, Helmholtz Institute for Pharmaceutical Research Saarland (HIPS), Saarland University Campus, Saarbrücken, Germany

[2]Department of Pharmacy, Saarland University, Saarbrücken, Germany

[3]Helmholtz Centre for Infection Research (HZI), Braunschweig, Germany

[4]DZIF-German Center for Infection Research, partner site Hannover-Braunschweig, Braunschweig, Germany

[5]Faculty of Pharmacy, University of Khartoum, Khartoum, Sudan

[6]Helmholtz International Lab for Anti-Infectives, Saarbrücken, Germany

[7]Parasite Chemotherapy Unit, Swiss Tropical and Public Health Institute, Allschwil, Switzerland

[8]Faculty of Science, University of Basel, Basel, Switzerland

## AUTHOR ORCIDs

Sebastian Walesch http://orcid.org/0000-0002-1101-0939
Daniel Krug http://orcid.org/0000-0002-1042-5665
Rolf Müller http://orcid.org/0000-0002-1042-5665

## AUTHOR CONTRIBUTIONS

Sebastian Walesch, Conceptualization, Data curation, Investigation, Methodology, Visualization, Writing – original draft, Writing – review and editing | Ronald Garcia, Investigation, Writing – review and editing | Abdelhalim B. Mahmoud, Writing – original draft, Writing – review and editing | Fabian Panter, Conceptualization, Writing – review and editing | Sophie Bollenbach, Investigation | Pascal Mäser, Writing – review and editing | Marcel Kaiser, Data curation, Investigation, Writing – review and editing | Daniel Krug, Supervision, Writing – review and editing | Rolf Müller, Funding acquisition, Supervision, Writing – review and editing

## DATA AVAILABILITY

The identified angiolam biosynthetic gene cluster was deposited with GenBank under accession number PP073370.

## ADDITIONAL FILES

The following material is available online.

### Supplemental Material

**Supplemental material (Spectrum03689-23-s0001.pdf).** Supplemental methods and results.

### Open Peer Review

**PEER REVIEW HISTORY (review-history.pdf).** An accounting of the reviewer comments and feedback.

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
