## [Reviewer comments · Microbiology Spectrum]

Microbiology Spectrum

New myxobacteria of the *Myxococcaceae* clade produce angiolams with antiparasitic activities

Sebastian Walesch, Ronald Garcia, Abdelhalim Mahmoud, Fabian Panter, Sophie Bollenbach, Pascal Mäser, Marcel Kaiser, Daniel Krug, and Rolf Müller

Corresponding Author(s): Rolf Müller, Helmholtz-Institut für Pharmazeutische Forschung Saarland

Review Timeline:

Submission Date:	October 16, 2023
Editorial Decision:	November 17, 2023
Revision Received:	December 21, 2023
Accepted:	January 2, 2024

Editor: Xiaoyu Tang

Reviewer(s): Disclosure of reviewer identity is with reference to reviewer comments included in decision letter(s). The following individuals involved in review of your submission have agreed to reveal their identity: Fei Xu (Reviewer #1)

Transaction Report:

DOI: <https://doi.org/10.1128/spectrum.03689-23>

Re: Spectrum03689-23 (New myxobacteria of the Myxococcaceae clade produce angiols with antiparasitic activities)

Dear Prof. Rolf Müller:

Thank you for the privilege of reviewing your work. Below you will find my comments, instructions from the Spectrum editorial office, and the reviewer comments.

Revision Guidelines

Sincerely,
Xiaoyu Tang
Editor
Microbiology Spectrum

Reviewer #1 (Comments for the Author):

This manuscript entitled "New myxobacteria of the Myxococcaceae clade produce angiols with antiparasitic activities" reported the discovery of a series of novel angiols analogs, structure elucidation, bioactivities and their proposed biosynthetic pathway. Considering the remarkable antiparasitic activities of these analogs, this work provides good drug leads and the combination biosynthesis potential of novel angiols.

1. One interesting point is analogs D1, D2 and F, which indicated the last two AT domain could load both methylmalonyl-CoA and ethylmalonyl-CoA. This is rare in normal PKS assembly line, thus the AA sequence alignment of ATs should be provided to see the differences key amino acid residue.

2. The main draft lacked the whole gene cluster arrow figure and annotation of gene function, should be added.

Reviewer #2 (Comments for the Author):

The manuscript describes 5 novel angiolam derivatives derived from new myxobacteria with improved bioactivities against *P. falciparum*. Also, the authors uncovered the biosynthetic gene cluster of angiolams including a potentially rare enzymatic dehydrogenation of the starter unit. This work reveals the diversity of natural products has been largely underappreciated in myxobacteria, and the distinct advantage of computational tools and suitable bioassays in natural products discovery. Therefore, this reviewer considers only a few suggestions for improvement as listed below.

1) Figure 1, I suggest replacing *Angiococcus disciformis* with *Pyxidicoccus fallax*, that will articulate clarify the correlation between angiolams and the genus *Pyxidicoccus*.

2) line 129, The EIC of which strain was shown in Figure 2A? MCy12716 or MCy12733?

3) Why does the EIC spectrum in Figure 2A show that the compound has multiple peaks, even for the major product angiolams A?

4) line 177, Figure 2B should be corrected to Figure 2C.

5) line 181, Figure 2A should be corrected to Figure 2B.

Reviewer #1 (Comments for the Author):

This manuscript entitled "New myxobacteria of the Myxococcaceae clade produce angiols with antiparasitic activities" reported the discovery of a series of novel angiols analogs, structure elucidation, bioactivities and their proposed biosynthetic pathway. Considering the remarkable antiparasitic activities of these analogs, this work provides good drug leads and the combination biosynthesis potential of novel angiols.

1. One interesting point is analogs D1, D2 and F, which indicated the last two AT domain could load both methylmalonyl-CoA and ethylmalonyl-CoA. This is rare in normal PKS assembly line, thus the AA sequence alignment of ATs should be provided to see the differences key amino acid residue.
2. The main draft lacked the whole gene cluster arrow figure and annotation of gene function, should be added.

Response:

Thank you for your kind assessment and comments.

- 1. The AA sequence alignments of all AT domains in the ang BGC can be found in the supplemental material. As part of the discussion regarding the incorporation of either methylmalonyl-CoA or ethylmalonyl-CoA in modules 11 and 12, we have referenced the alignment figure in the supplemental material (Figure S4).*
- 2. Following the comparison of genes in the ang BGC with the genome of the closely related P. caerfyrdinensis CA032A that does not feature the core biosynthetic genes angA-angF but the surrounding genes, we decided to only display AngA-AngF in the gene cluster arrow figure. However, we have added a gene cluster arrow figure including genes ang1-ang6 upstream and ang7-ang8 downstream of angA-angF in the supplemental material (Figure S3). Moreover, the gene functions and closest relatives for all genes of the ang BGC can be found in the supplemental material (Table S 20-S 21).*

Reviewer #2 (Comments for the Author):

The manuscript describes 5 novel angiols derivatives derived from new myxobacteria with improved bioactivities against *P. falciparum*. Also, the authors uncovered the biosynthetic gene cluster of angiols including a potentially rare enzymatic dehydrogenation of the starter unit. This work reveals the diversity of natural products has been largely underappreciated in myxobacteria, and the distinct advantage of computational tools and suitable bioassays in natural products discovery. Therefore, this reviewer considers only a few suggestions for improvement as listed below.

- 1) Figure 1, I suggest replacing *Angiococcus disciformis* with *Pyxidicoccus fallax*, that will articulate clarify the correlation between angiols and the genus *Pyxidicoccus*.
- 2) line 129, The EIC of which strain was shown in Figure 2A? MCy12716 or MCy12733?

- 3) Why does the EIC spectrum in Figure 2A show that the compound has multiple peaks, even for the major product angiolams A?
- 4) line 177, Figure 2B should be corrected to Figure 2C.
- 5) line 181, Figure 2A should be corrected to Figure 2B.

Response:

Thank you for your kind assessment and comments. We have updated the manuscript according to your suggestions.

- 1. We have updated the figure, it now states Pyxidicoccus fallax An d30.*
- 2. The EICs presented in Figure 2A belong to angiolam derivatives produced by MCy12733. We have specified this in the legend of Figure 2.*
- 3. Based on the β -keto-amide moiety of angiolams at positions 15-17 we hypothesise a keto-enol tautomerism at positions C-15 and C-16 leading to the observed elution pattern on RP C-18 columns.*
- 4. & 5. We have corrected the figure descriptions.*

Re: Spectrum03689-23R1 (New myxobacteria of the *Myxococcaceae* clade produce angiolsams with antiparasitic activities)

Dear Prof. Rolf Müller:

Your manuscript has been accepted, and I am forwarding it to the ASM production staff for publication. Your paper will first be checked to make sure all elements meet the technical requirements. ASM staff will contact you if anything needs to be revised before copyediting and production can begin. Otherwise, you will be notified when your proofs are ready to be viewed.

Sincerely,
Xiaoyu Tang
Editor
Microbiology Spectrum